# Differential regulation of breast cancer bone metastasis by PARP1 and PARP2

Hao Zuo [1], Dengbao Yang [1], Qiwen Yang [1], Haidong Tang [2], Yang-Xin Fu [2] & Yihong Wan [1,3 ✉]

PARP1 and PARP2 dual inhibitors, such as olaparib, have been recently FDA approved for the treatment of advanced breast and ovarian cancers. However, their effects on bone mass and bone metastasis are unknown. Here we show that olaparib increases breast cancer bone metastasis through PARP2, but not PARP1, specifically in the myeloid lineage, but not in the cancer cells. Olaparib treatment or PARP1/2 deletion promotes osteoclast differentiation and bone loss. Intriguingly, myeloid deletion of PARP2, but not PARP1, increases the population of immature myeloid cells in bone marrow, and impairs the expression of chemokines such as CCL3 through enhancing the transcriptional repression by β-catenin. Compromised CCL3 production in turn creates an immune-suppressive milieu by altering T cell subpopulations. Our findings warrant careful examination of current PARP inhibitors on bone metastasis and bone loss, and suggest cotreatment with CCL3, β-catenin inhibitors, anti-RANKL or bisphosphonates as potential combination therapy for PARP inhibitors.

[1] Department of Pharmacology, The University of Texas Southwestern Medical Center, Dallas, TX 75390, USA. [2] Department of Pathology, The University of Texas Southwestern Medical Center, Dallas, TX 75390, USA. [3] Simmons Cancer Center, The University of Texas Southwestern Medical Center, Dallas, TX 75390, USA. ✉email: yihong.wan@utsouthwestern.edu

Bone metastasis is a frequent, debilitating, and essentially incurable cancer complication. More than 70% of patients with advanced breast cancer have bone metastases. They suffer from severe bone pain, pathological fractures, life-threatening hypercalcemia, limited mobility, and increased mortality. During bone metastasis, cancer cells and osteoclasts form a vicious cycle so that cancer cells promote osteoclast differentiation and osteoclasts in turn facilitate cancer-cell seeding and proliferation in bone[1–4]. Blocking osteoclast function by bisphosphonates or RANKL antibody are currently FDA-approved strategies to attenuate metastatic bone disease. However, these drugs are only at best palliative and do not improve overall patient survival. As such, bone metastases remain incurable and better therapeutics is urgently needed, which requires new understanding of cellular and molecular mechanisms.

Moreover, the clinical development of many cancer drugs is mainly based on shrinking primary tumors, whereas their effects on bone metastasis are still underexplored. Poly ADP-ribosylation is a post-translational protein modification in which certain members of poly (ADP-ribose) polymerase (PARP), such as PARP1 and PARP2, generate poly ADP-ribose (PAR) chains on target proteins by transferring ADP ribosyl groups from $NAD^+$ onto the side chain of residues. PARP1 and PARP2 both regulate DNA repair[5,6] and transcription;[7] yet they also have distinct functions[8,9]. PARP inhibitors cause synthetic lethality in BRCA-mutated cancer cells from defective DNA damage repair[10,11]. In 2014, an inhibitor of PARPs, olaparib, was approved by the FDA for treating relapsed BRCA-defective ovarian cancer. In 2018, olaparib was approved for the treatment of germline BRCA-mutated metastatic breast cancer. Many other PARP1/2 dual inhibitors are also approved or in clinical trials. Despite the clinical efforts on PARP inhibitors, little is known about the potential roles of PARPs in metastatic bone diseases or the impact of PARP inhibitors on bone metastasis.

## Results

### Increased breast cancer bone metastasis by olaparib or PARP2 deficiency.
To determine whether PARP inhibitors alter breast cancer bone metastasis, we performed intracardiac injection of luciferase-labeled breast cancer cells into mice and then treated them with olaparib. A bone-met-prone human breast cancer cell line MDA-MB-231-derived 1833 cells was xenografted into nude mice, as described by our and other groups[12,13]. Bone-met-prone mouse mammary tumor cell line 4T1-derived 4T1.2 (ref. [13]) or MMTV-PyMT mice-derived Py8119 (ref. [14]) was allografted into BALB/c or C57BL/6 (B6) mice, respectively, which represent syngeneic mouse models. The results show that nude mice or BALB/c mice treated with olaparib developed significantly more bone metastases from 1833 cells or 4T1.2 cells, respectively, compared with those treated with DMSO vehicle control, quantified by bioluminescence imaging (BLI; Fig. 1a, b). X-ray imaging revealed significantly more osteolytic lesions at the sites of bone metastases in mice treated with olaparib than in the control mice (Supplementary Fig. 1a). H&E staining also confirmed bone metastases (Supplementary Fig. 1b).

Next, we investigated which cell types mediate olaparib effects on bone metastasis. We first examined the individual roles of PARP1 and PARP2 in cancer cells. PARP2 knock down (KD) in 1833 cells resulted in significantly more bone metastases and osteolytic lesions compared with control cells (Supplementary Fig. 1c–e). Conversely, PARP2 overexpression in 1833 cells reduced bone metastasis and bone lesions (Supplementary Fig. 1f, g). Consistent with these observations, bone metastasis from Py8119 cells in B6 mice was also increased by cancer cell PARP2KD (Supplementary Fig. 1h). In contrast, PARP1KD in cancer cells did

not affect bone metastasis in either nude mice or B6 mice (Supplementary Fig. 1i, j).

Osteoclasts are bone-resorbing cells derived from the myeloid lineage. They form a vicious cycle with cancer cells that propels skeletal metastasis[1–4]. We found that the serum levels of a bone resorption marker CTX-1 (carboxy-terminal telopeptides of type I collagen) were significantly elevated in mice injected with PARP2KD cancer cells compared with control cancer cells (Supplementary Fig. 1k, l), indicating increased osteoclast functions in these mice. Osteoclast precursors cocultured with PARP2KD cancer cells showed increased expression of osteoclast marker gene cathepsin K (CTSK; Supplementary Fig. 1m). PARP2KD cancer cells exhibited higher RANKL expression, but lower osteoprotegerin (OPG) expression, leading to an increased RANKL/OPG ratio (Supplementary Fig. 1n), thus facilitating osteoclast differentiation and bone resorption (Supplementary Fig. 1o). In vitro analysis showed that PARP2KD cancer cells exhibited a slight but significant increase in proliferation and migration compared to control cancer cells (Supplementary Fig. 1p, q). Surprisingly, although PARP2KD in cancer cells was sufficient to increase bone metastasis, PARP2KD in cancer cells did not abolish the bone-met-enhancing effects of olaparib treatment as olaparib could further exacerbate bone metastasis from PARP2KD cancer cells (Fig. 1c). This data suggests that PARP2 in cancer cells only plays minor roles and other cell types in the bone environment may be involved.

To determine the effect of PARP deletion in the entire cancer environment on bone metastasis, we examined PARP1 or PARP2 global knockout (KO) mice. Nude mice and B6 mice with global PARP2 deletion developed more bone metastases than their wild-type (WT) littermate controls, after intracardiac injection with 1833 cells or Py8119 cells, respectively (Fig. 1d, Supplementary Fig. 2a). In contrast, PARP1KO nude mice or B6 mice did not show altered bone metastasis (Supplementary Fig. 2b, c). These results indicate that PARP2, but not PARP1, in the cancer environment suppresses bone metastasis.

In light of the important roles of osteoclast in bone metastasis, we generated and analyzed mice with PARP2 or PARP1 conditional knockout (CKO) in the myeloid/osteoclast lineage using lysozyme-Cre (LyzM-Cre) driver. Bone metastasis was significantly increased in PARP2 conditional KO (PARP2CKO) mice (Fig. 1e), but not in PARP1CKO mice (Supplementary Fig. 2d), compared with corresponding littermate control mice. Furthermore, the ability of olaparib to exacerbate bone metastasis of Py8119 cells in WT mice was completely abolished in PARP2CKO mice (Fig. 1f), pinpointing that olaparib-induced bone metastasis is predominantly mediated by PARP2 in the myeloid lineage and less by PARP2 in the cancer cells.

We also examined a spontaneous breast cancer model MMTV-PyMT mice with global PARP2KO or PARP2CKO in mammary gland or myeloid lineage using MMTV-Cre or LyzM-Cre, respectively. Tumor onset was significantly delayed in PARP2 global KO and mammary KO mice, but not in myeloid KO mice, as compared to their respective WT controls (Supplementary Fig. 2e–g). However, tumor growth rate (days from tumor onset to maximal tumor volume limit) was unaltered (Supplementary Fig. 2h, i). In line with this observation, neither PARP1/2 KD in cancer cells nor PARP1/2 deletion in mice affected tumor growth in orthotopic mammary fat pad tumor implantation models (Supplementary Fig. 2j–m). Lung, although has a quite different microenvironment from bone, is another common metastasis site of breast cancer. We found that spontaneous lung metastasis was significantly reduced in PARP2KO MMTV-PyMT mice, as well as in PARP2KD or PARP1KD cancer cell-implanted mice (Supplementary Fig. 2n–p), whereas bone metastasis was rarely observed in the MMTV-PyMT and Py8119 mammary

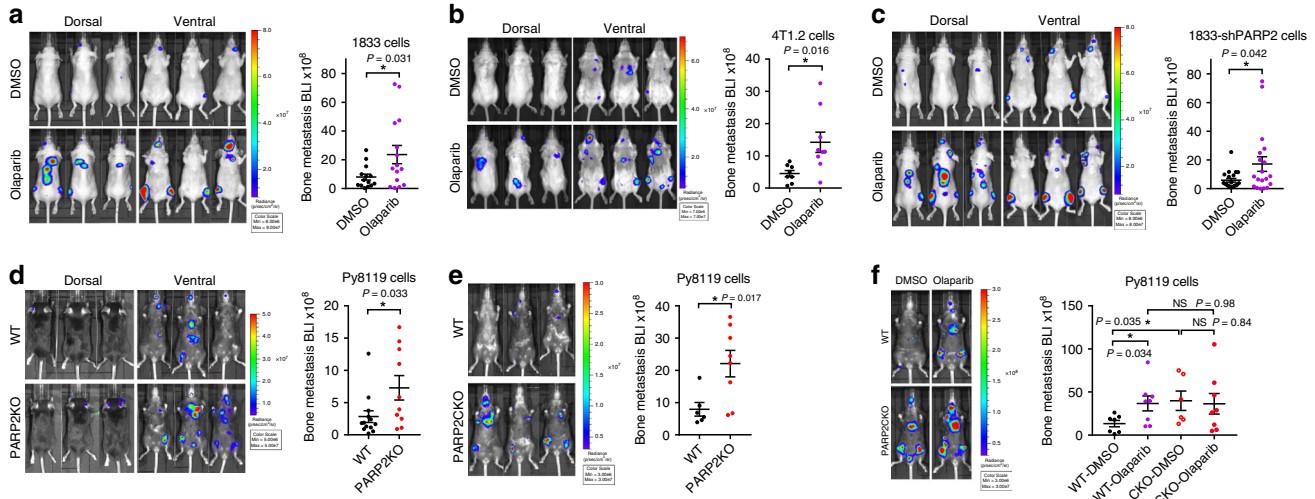

**Fig. 1 PARP2 deficiency increases breast cancer bone metastasis.** Olaparib treatment increased bone metastasis of MDA231-BoM-1833 cells in nude mice ($n = 14$, **a**) and 4T1.2 cells in BALB/c mice ($n = 9$, **b**). **c** Bone metastases of PARP2KD 1833 cells after olaparib treatment ($n = 20$) were more than DMSO control ($n = 15$). **d** Bone metastasis of Py8119 cells was increased in PARP2KO mice ($n = 10$) compared with B6 wild-type (WT) control mice ($n = 13$). **e** PARP2CKO mice ($n = 8$) developed more bone metastases of Py8119 cells than B6 WT mice ($n = 6$). **f** Treatment of olaparib increased bone metastasis of Py8119 cells in B6 WT mice ($n = 8$), but not in PARP2CKO mice ($n = 8$), compared with DMSO-treated WT ($n = 7$) or PARP2CKO ($n = 6$) mice, respectively. **a–f** Left, bioluminescence imaging (BLI) images; right, quantification of BLI signals. Data represent mean ± SEM, $^*P < 0.05$; NS nonsignificant. Two-sided Student's $t$-test was used to calculate statistical difference. Source data are provided as a Source Data file.

implantation models. Together, our data indicate that PARP2 deficiency suppresses tumor onset and lung metastasis yet increases bone metastasis without affecting primary tumor growth, suggesting that the microenvironment in bone is specifically responsible for the olaparib induction of bone metastasis.

**Enhanced osteoclast differentiation and bone loss by olaparib or deletion of PARP1/2.** We next examined how olaparib and PARP2 function in the myeloid lineage to modulate bone metastasis. First, we determined the effects of olaparib on osteoclastogenesis and bone mass. Micro-computed tomography (μCT) imaging revealed that olaparib treatment in mice for 5 weeks led to reduced trabecular bone volume and trabecular number with increased trabecular space and structure model index (SMI), which quantifies the relative amount of plates (SMI = 0, strong) and rods (SMI = 3, fragile; Fig. 2a, b). This low bone mass phenotype was due to significantly augmented osteoclastogenesis shown by the increased osteoclast number and osteoclast surface (Fig. 2c). Ex vivo bone marrow osteoclast differentiation, which was induced with RANKL and further stimulated with rosiglitazone as we previously described[13,15], was enhanced by olaparib, illustrated by the larger number and size of multinucleated cells stained positive for the osteoclast marker tartrate-resistant acid phosphatase (TRAP), as well as the higher TRAP mRNA induction (Fig. 2d). On the other hand, PARP activation using PDD 00017273, an inhibitor of the PAR-degrading enzyme PARG, suppressed osteoclastogenesis (Supplementary Fig. 3a). PARP2KO bone marrow precursors showed enhanced osteoclast differentiation yet were still sensitive to the pro-osteoclastogenic effects of olaparib (Fig. 2d), indicating that both PARP1 and PARP2 impede osteoclast differentiation.

We next tested whether PARP2 deficiency in the myeloid/osteoclast lineage is sufficient to enhance osteoclastogenesis and cause low bone mass. μCT revealed that LyzM-Cre-mediated PARP2CKO mice also had lower trabecular volume, trabecular number, trabecular thickness, higher trabecular separation, decreased bone mineral density (BMD), and increased SMI

(Fig. 2e, f). Moreover, PARP2CKO mice also exhibited lower cortical bone volume and thickness (Supplementary Fig. 3b). The number and surface of osteoclasts in bone sections were significantly elevated in PARP2CKO mice (Fig. 2g). PARP2CKO cultures also showed enhanced osteoclast differentiation (Fig. 2h). PARP2 global KO mice had similar bone phenotypes as PARP2CKO mice, including low bone mass, more osteoclasts in bone and culture (Supplementary Fig. 3c–f). Consistent with a previous report that PARP1 global KO mice have osteopenia due to increased osteoclastogenesis[16], we found that our PARP1CKO mice also displayed increased osteoclast differentiation and decreased bone mass (Supplementary Fig. 3b, g–j). PARP inhibitor has been reported to suppress bone-forming osteoblasts[17]. We also observed impaired bone formation in olaparib-treated mice, PARP1 global KO mice, and PARP2 global KO mice (Supplementary Fig. 3k), indicating that increased osteoclastogenesis is not due to elevated osteoblastogenesis.

Deficiency of PARP1 and PARP2 had comparable effects on increasing the expression of osteoclast marker genes and receptor RANK (Supplementary Fig. 4a, b). PARP1 cleavage is required for osteoclast differentiation as PARP1 uncleavable mutant decreases osteoclast number and increases bone mass[18]. In WT bone marrow cultures, both PARP1 and PARP2 proteins were cleaved during osteoclast differentiation (Supplementary Fig. 4c–f). To pinpoint the mechanisms underlying PARP2 regulation of osteoclastogenesis, we generated three PARP2 mutants: D61E for loss of caspase 3 cleavage[19], D187E for loss of caspase 8 cleavage[20], and E534A for loss of catalytic activity. We found that only D61E mutant was able to suppress osteoclast differentiation when overexpressed in RAW264.7 cells, a murine macrophage cell line that can differentiate into osteoclasts under RANKL treatment (Supplementary Fig. 4g, h). Together, the results from our study and a previous report[18] indicate that cleavage of both PARP1 and PARP2 are required for osteoclast differentiation. These findings uncover bone protective roles of PARP1 and PARP2, and suggest potential bone loss side effects from PARP inhibitor cancer drugs. However, given that the bone-met-enhancing effects of PARP deficiency or PARP inhibition were mainly mediated by myeloid PARP2, but not PARP1 (Fig. 1),

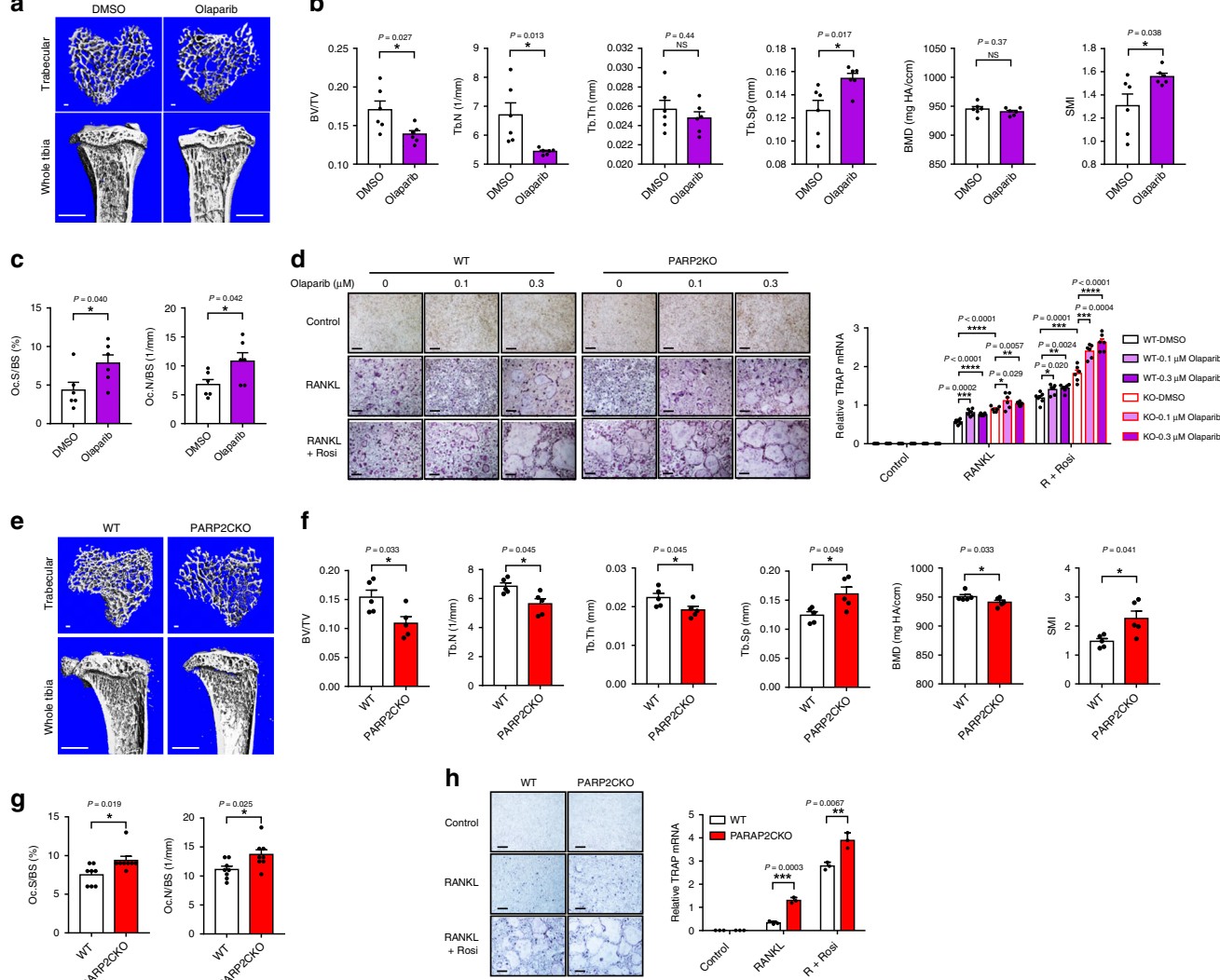

**Fig. 2 Olaparib or PARP2 deletion reduces bone mass and promotes osteoclast differentiation. a, b** Micro-computed tomography (μCT) of the tibiae from B6 mice treated with DMSO or olaparib (2-month-old male, 5-week daily treatment, $n = 6$). **a** Images of the trabecular bone of the tibial metaphysis (top; scale bar, 100 μm) and the entire proximal tibia (bottom; scale bar, 1 mm). **b** Trabecular bone parameters showed reduced BV, Tb.N, and increased Tb.Sp, SMI of olaparib-treated B6 mice compared with DMSO-treated mice. BV/TV bone volume/tissue volume ratio, Tb.N trabecular number, Tb.Th trabecular thickness, Tb.Sp trabecular separation, BMD bone mineral density, SMI structure model index. **c** Histomorphometry of TRAP-stained bone sections showed increased osteoclasts in the distal femur from B6 mice treated with olaparib compared with DMSO ($n = 6$). Oc.S osteoclast surface, Oc.N osteoclast number. **d** Treatment with 0.1 or 0.3 μM olaparib increased in vitro osteoclast differentiation of WT or PARP2KO cultures ($n = 6$). Left, TRAP staining (scale bar, 25 μm); right, TRAP expression. R RANKL, Rosi Rosiglitazone. **e, f** μCT of the trabecular bone in tibiae shows that PARP2CKO mice had less BV, Tb.N, Tb.Th, BMD, and more Tb.Sp, SMI than WT mice (2-month-old male, $n = 5$). **g** Histomorphometry of TRAP-stained bone sections of the distal femur shows that PARP2CKO mice had increased osteoclasts compared with WT mice (2-month-old male, $n = 8$). **h** PARP2CKO cultures showed increased osteoclast differentiation ($n = 3$). Left, TRAP staining (scale bar, 25 μm); right, TRAP expression. Data represent mean ± SEM, *$P < 0.05$, **$P < 0.01$, ***$P < 0.001$, ****$P < 0.0001$; NS nonsignificant. Two-sided Student's $t$-test was used to calculate statistical difference. Source data are provided as a Source Data file.

these observations suggest that osteoclast may not be the key cell type in the bone environment responsible for PARP2-specific regulation of bone metastasis.

**Immature myeloid cell accumulation, CCL3 deficiency, and altered T helper cells by olaparib or PARP2 deletion.** To delineate which cell type in the myeloid lineage, other than osteoclasts, confers the differential regulation of bone metastasis by PARP1 and PARP2, we performed flow cytometry analyses (FACS) of bone marrow cells isolated from mice receiving cancer cells via intracardiac injection. Interestingly, of all the populations

we examined, CD11b + Gr1+ immature myeloid cells (IMCs) were the only cell type that exhibited similar significant changes by both PARP2CKO and olaparib treatment (Fig. 3a, Supplementary Fig. 5a–c). Olaparib also increased IMCs in BALB/c mice injected with 4T1.2 cells (Fig. 3b). In contrast, PARP1CKO mice had a similar population of IMCs as WT mice after intracardiac injection (Supplementary Fig. 5d). Increased IMCs in the marrow upon PARP2 deletion or inhibition may enhance bone metastasis through IMCs' capabilities in supporting tumor cell growth and immunosuppression[21]. As LyzM-Cre targets the entire myeloid lineage and PARP1/2 are cleaved after RANKL treatment (Supplementary Fig. 4c–f, i), we focused on the functions of PARP1/2

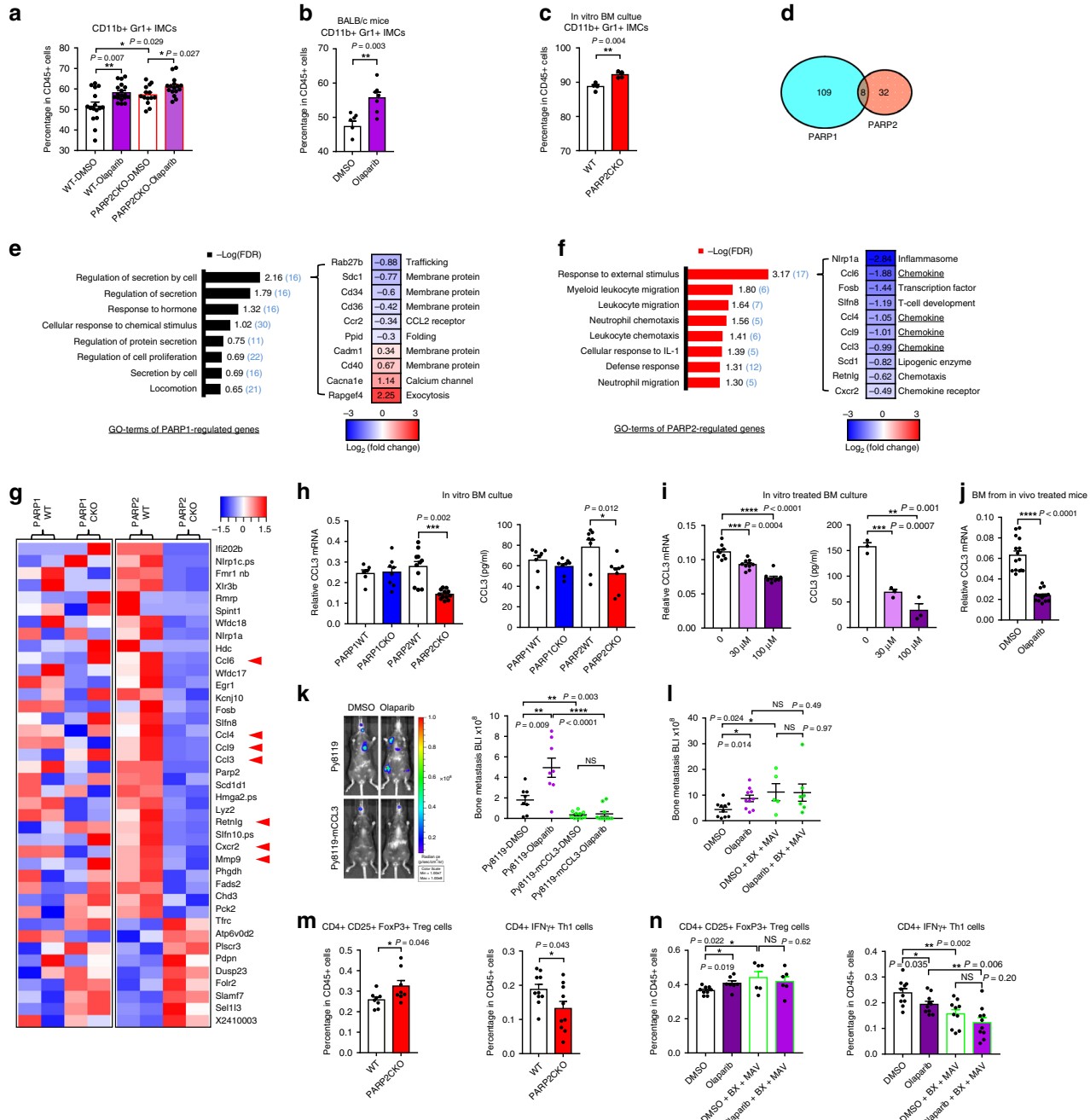

**Fig. 3 PARP2 deficiency increases IMCs and decreases CCL3 expression. a** Immature myeloid cells (IMCs) were increased by both olaparib and PARP2CKO shown by FACS analyses of bone marrow cells from mice intracardiacally injected with Py8119 cells (*n* = 16). **b** Olaparib increased IMCs in bone marrow of BALB/c mice intracardiacally injected with 4T1.2 cells (*n* = 7). **c** PARP2CKO increased IMCs of bone marrow cultures after 3 days of M-CSF treatment (*n* = 4). **d–g** RNA-seq analyses of differential gene expressions in IMCs of PARP1CKO or PARP2CKO (*n* = 2). **d** Diagram representation of overlap between PARP1 and PARP2 regulated genes. **e, f** Gene ontology terms enriched for PARP1 **e** or PARP2 **f** regulated genes and their top regulated genes. **g** Heat maps of expression levels for PARP2 regulated genes. Arrows, genes related to leukocyte migration. **h** RT-qPCR and ELISA analyses showed that CCL3 was reduced only in PARP2CKO IMCs (*n* = 8). **i** CCL3 mRNA (left) or protein (right) in IMCs was suppressed by the treatment of 30 μM or 100 μM olaparib for 1 day (*n* = 3). **j** CCL3 mRNA was reduced in the bone marrow cells of olaparib-treated B6 mice (2-month-old female, 5-week daily treatment, *n* = 15). **k** Bone metastasis of Py8119-mCCL3 cells in mice treated with DMSO or olaparib (*n* = 10) was lower than Py8119 cells (*n* = 8). **l** Bone metastasis of Py8119 cells in mice treated with CCR1 antagonist, BX-471 (BX), plus CCR5 antagonist, Maraviroc (MAV; *n* = 7) was exacerbated compared with DMSO group (*n* = 10). **m** PARP2CKO increased regulatory T cells (Treg; *n* = 8) and reduced type 1 T helper cells (Th1; *n* = 10) in bone marrow of mice intracardiacally injected with Py8119 cells. **n** Both olaparib and CCR1/5 antagonists increased Treg (*n* = 6) and reduced Th1 cells (*n* = 10) in bone marrow from mice in **l**. Data represent mean ± SEM, *P < 0.05, **P < 0.01, ***P < 0.001, ****P < 0.0001; NS nonsignificant. Two-sided Student's *t*-test was used to calculate statistical difference. Source data are provided as a Source Data file.

in IMCs expanded ex vivo from bone marrow progenitors (Supplementary Fig. 4i). FACS analyses showed that the majority of these cells were CD11b + Gr1+ IMCs, the percentage of which was increased by PARP2CKO (Fig. 3c), further supporting a role of IMC in mediating the PARP2-specific regulation of bone metastasis.

We performed expression profiling by RNA-seq to compare IMCs from PARP1CKO and PARP2CKO mice with their corresponding littermate controls. The results showed that genes regulated by PARP1 and PARP2 were largely distinct with only a small subset overlapping (Fig. 3d). Gene ontological analyses with DAVID (the Database for Annotation, Visualization and Integrated Discovery)[22] revealed that PARP1 regulated genes enrich terms related to secretion (Fig. 3e), whereas PARP2 regulated genes enrich terms related to response to external stimulus and leukocyte migration (Fig. 3f). PARP2 deficiency downregulated the expression of immune chemokines, such as CCL3, CCL4, CCL6, and CCL9 (Fig. 3f, g, Supplementary Table 1). In contrast, these genes were unaffected by PARP1 deficiency (Fig. 3g, Supplementary Table 1). This differential gene regulation by PARP2 and PARP1 was validated by RT-qPCR (Fig. 3h, Supplementary Fig. 5e). Olaparib was also able to decrease CCL3 mRNA and protein levels both in vitro and in vivo (Fig. 3i, j). Moreover, olaparib reduced CCL4, 6, and 9 in vitro but only lowered CCL4 expression in vivo (Supplementary Fig. 5f, g).

We tested the effect of rescuing CCL3 level in the bone metastatic microenvironment by CCL3 overexpression in breast cancer cells (Supplementary Fig. 6a). Olaparib-exacerbated bone metastasis was abolished by mCCL3 overexpression in both Py8119 and 1833 cells in B6 and nude mice, respectively (Fig. 3k, Supplementary Fig. 6b). Moreover, elevating CCL3 level was sufficient to suppress basal bone metastasis in immunocompetent B6 mice, but not in athymic nude mice (Fig. 3k, Supplementary Fig. 6b). These results indicate that CCL3 regulation of bone metastasis may be mediated by both T cells and innate immune cell types. CCL3 is a critical chemokine for differentiation, chemotaxis, and activation of immune cells, including macrophages, dendritic cells, natural killer cells, and T cells[23], we found that CCL3 attracted the migration of all the bone marrow immune cell types, most of which were blocked by antagonists of CCL3 receptor CCR1/5 (Supplementary Fig. 6c). As a complementary loss-of-function in vivo approach, we treated mice with CCR1/5 antagonists to inhibit the endogenous CCL3 signaling; as a result, bone metastasis was exacerbated and olaparib could no longer further increase bone metastasis (Fig. 3l), confirming a physiologically significant role of CCL3 signaling in suppressing bone metastasis and mediating olaparib regulation. As T cells may be responding to CCL3 during bone metastasis, we analyzed T cell subpopulations with FACS. Both PARP2CKO and olaparib increased the population of immune-suppressive CD25 + FoxP3+ regulatory T cells (Treg) and reduced the population of immune-activating IFNγ + type 1 T helper cells (Th1; Fig. 3m, n). The effect of olaparib on Treg and Th1 was also impaired by CCR1/5 antagonists (Fig. 3n), supporting that it is CCL3 mediated. Olaparib also increased the population of type 2 T helper cells (Th2) and T helper 17 cells (Th17), but these effects were likely not mediated by PARP2 or CCL3 because they were not observed upon PARP2CKO or CCR1/5 inhibition (Supplementary Fig. 6d, e). Interestingly, olaparib was still able to increase IMC population after CCL3 replenishment or CCL3 signaling blockade, suggesting that reduced CCL3 expression is a downstream event of IMC accumulation (Supplementary Fig. 6f, g). All these results suggest that PARP2CKO and olaparib cause IMC accumulation in bone, impair CCL3 levels, and convert the bone microenvironment to immunosuppressive by altering the balance of Treg and Th1 cells.

**Enhanced repression of CCL3 transcription by β-catenin after PARP2 deletion.** We next investigated the molecular mechanism for how PARP2 enhances CCL3 expression. Chromatin immunoprecipitation (ChIP) assay showed that PARP2 bound to CCL3 proximal promoter in both IMCs and cancer cells (Fig. 4a). The ChIP signals were PARP2 specific as they were significantly reduced in PARP2KD or PARP2CKO cells (Fig. 4a). In contrast, PARP1 did not bind to CCL3 promoter (Supplementary Fig. 7a). Consistent with the genetic analysis using PARP2 depletion, pharmacological PARP inhibition via olaparib treatment also diminished CCL3 promoter binding with PARP2 (Fig. 4b), but not PARP1 (Supplementary Fig. 7b). We cloned different regions of CCL3 promoter upstream of a luciferase reporter (Fig. 4c). Transient transfection and reporter assay showed that PARP2KD, but not PARP1KD, significantly impaired the luciferase output driven by mouse and human CCL3 promoters (Fig. 4c, Supplementary Fig. 7c). The level of H3K4me3, a histone mark for active transcription, was also significantly reduced at CCL3 promoter in PARP2-deficient cells (Supplementary Fig. 7d).

PARPs may regulate gene expression in conjunction with transcription factors[7]. Because CCL3 transcription is downregulated by β-catenin and upregulated by NFκB (refs. [24,25]), we next tested the effects of PARP deficiency on CCL3 promoter association with these two transcription factors. ChIP assay showed that β-catenin binding at CCL3 promoter was significantly augmented by the KD of PARP2 but not by PARP1 in cancer cells (Fig. 4d, Supplementary Fig. 7e). Consistently with this observation, β-catenin binding at CCL3 promoter was also enhanced in PARP2CKO IMCs (Fig. 4e). In contrast, p65/NFκB binding at CCL3 promoter was reduced by both PARP1 and PAR2 deficiency (Supplementary Fig. 7f). These results suggest that differential regulation of bone metastasis by PARP2 and PARP1 may be mediated by their distinct impact on β-catenin binding and suppression of CCL3 promoter. Indeed, TOP/FOP flash reporter assay revealed that β-catenin transcription activity is enhanced by the depletion of PARP2 but not PARP1, confirming a PARP2-specific β-catenin suppression (Fig. 4f, Supplementary Fig. 7g). Moreover, the expression of cyclin D1, a known β-catenin target gene, was elevated in PARP2-deficient cells (Supplementary Fig. 7h). It has been reported that β-catenin suppresses the expression of CCL3 (ref. [25]). We have indeed found consensus T cell factor (TCF) binding sites CTTTG(A/T)(A/T) within 1 kb upstream of CCL3 transcription start site (Supplementary Fig. 7i). Our results showed that the association of β-catenin with the CCL3 promoter was enhanced by PARP2 deletion (Fig. 4d, e). However, in vitro PARylation assay showed that β-catenin was not a substrate of either PARP1 or PARP2 (Supplementary Fig. 7j). Since both PARP2 and β-catenin can associate with the proximal CCL3 promoter, these findings indicate that olaparib binding to PARP2 releases PARP2 from CCL3 promoter (Fig. 4b), thus allowing β-catenin recruitment to CCL3 promoter.

We next performed rescue experiments in vitro and in vivo. Inhibition of β-catenin transcriptional activity by PRI-724 not only increased CCL3 mRNA and reduced cyclin D1 mRNA, but also abolish the effects of PARP2 deficiency in IMC cultures (Fig. 4g, Supplementary Fig. 7h). Cotreatment of olaparib with PRI-724 in mice also abrogated the bone metastasis induced by olaparib (Fig. 4h). These results support a key role of β-catenin over-activation in mediating the bone-met-enhancing effects of PARP2 deficiency and olaparib treatment. These findings also suggest that β-catenin inhibitors may be a potential combined therapy with PARP inhibitors for a better cancer treatment. Cancer patients with bone metastasis are often given anti-bone resorption agents, such as anti-RANKL antibody and bisphosphonates. We found that cotreatment with anti-RANKL

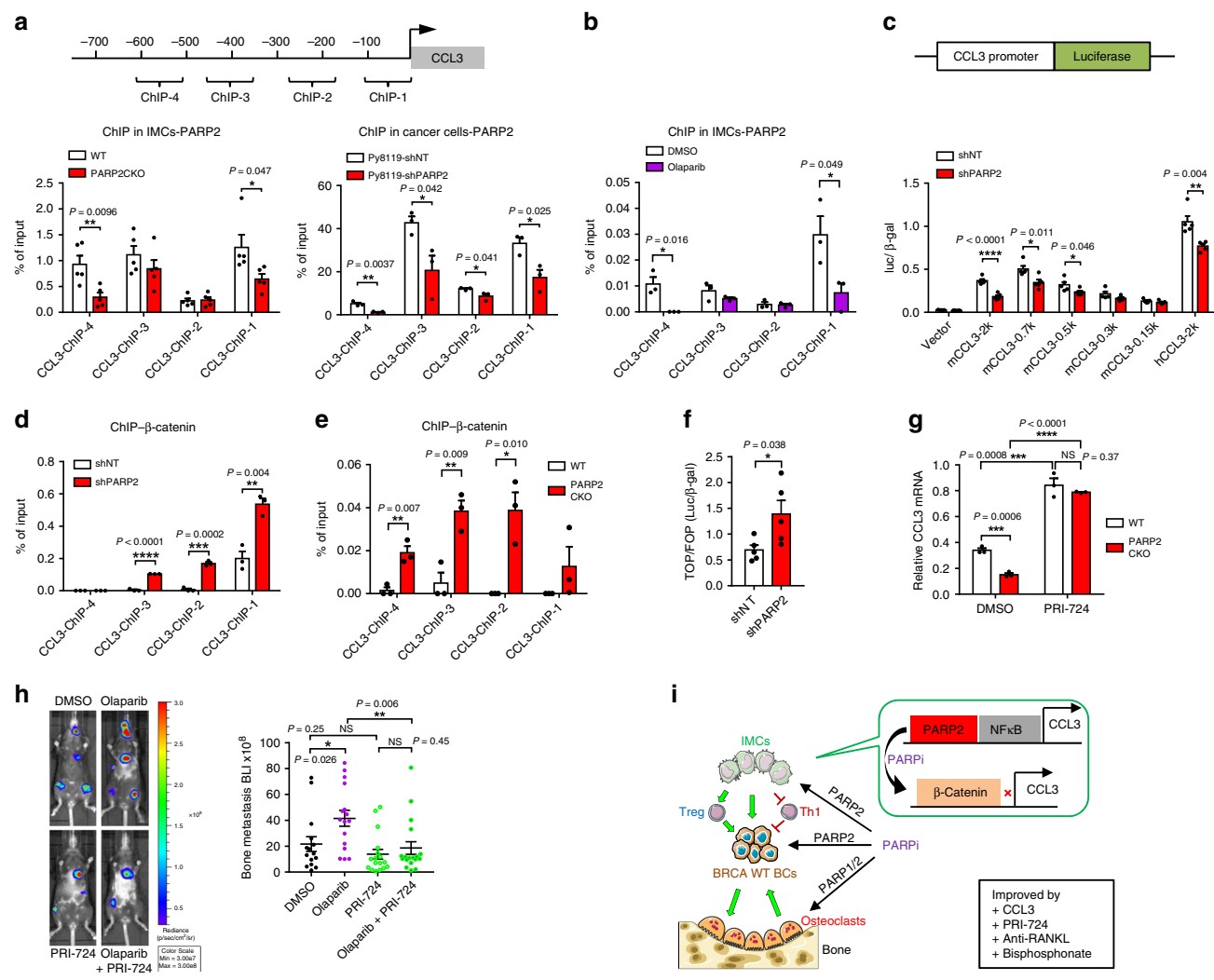

**Fig. 4 Inhibition of β-catenin increases PARP2-dependent CCL3 expression and prevents olaparib-induced bone metastasis. a** (Top) a diagram of primer locations at the promoter of CCL3 for chromatin immunoprecipitation (ChIP) analyses; (bottom) ChIP analyses showed that PARP2 associated with CCL3 promoter in IMCs (bottom left; $n = 5$) and Py8119 cells (bottom right; $n = 3$). **b** ChIP analyses showed that 3 μM olaparib-suppressed PARP2 binding to CCL3 promoter in IMCs ($n = 3$). **c** (Top) a diagram of CCL3 promoter-driven luciferase transcription in the CCL3-luc reporter system; (bottom) PARP2KD significantly impaired CCL3 promoter-driven luciferase expression ($n = 5$) when CCL3 promoter included >0.5k bp regions upstream of transcription start site. **d, e** In ChIP, β-catenin binding to CCL3 promoter was increased in PARP2KD Py8119 cells (**d**, $n = 3$) and PARP2CKO IMCs (**e**, $n = 3$). **f** TOP/FOP flash assay revealed that PARP2 negatively regulate β-catenin transcriptional activity ($n = 5$). **g** CCL3 expression was elevated in IMCs treated with 1 μM β-catenin inhibitor PRI-724 for 3 days ($n = 3$). **h** PRI-724 blocked olaparib-induced bone metastasis in B6 WT mice ($n = 16$). **i** A model for how olaparib regulates breast cancer bone metastasis. BCs breast cancer cells, PARPi PARP inhibitor. Data represent mean ± SEM, *$P < 0.05$, **$P < 0.01$, ***$P < 0.001$, ****$P < 0.0001$; NS nonsignificant. Two-sided Student's $t$-test was used to calculate statistical difference. Source data are provided as a Source Data file.

antibody or zoledronate (ZOL) significantly relieved olaparib-induced bone metastasis (Supplementary Fig. 7k).

We next tested whether breast cancer cell BRCA deficiency affects olaparib-induced bone metastasis. In vitro, olaparib showed similar inhibition of the growth and survival of BRCA-KD and BRCA-WT cancer cells, but it only significantly reduced the migration of BRCA-KD cancer cells but not BRCA-WT cancer cells (Supplementary Fig. 8a–d). In vivo, olaparib treatment also showed similar suppression of the primary tumor growth from BRCA-KD and BRCA-WT cancer cells (Supplementary Fig. 8e, f). In contrast to BRCA-WT cancer cells, bone metastasis from BRCA-KD cancer cells was reduced or unaffected by olaparib (Supplementary Fig. 8g, h). These results indicate that the impact of olaparib on bone metastasis may be dependent on contexts, such as the functional BRCA

status in cancer cells. However, among all clinical trials involving PARP inhibitor treatment of advanced ovarian and breast cancer, little attention was paid to the effects of PARP inhibitors on bone metastasis and no such data has been reported. Nevertheless, we summarized all currently available clinical data on bone health-related events in clinical trials (Supplementary Fig. 9a), and found that patients with PARP inhibitor treatment showed a trend of higher risk of bone fracture, serious bone, or musculoskeletal pain, as well as osteoporosis/osteonecrosis, compared with patients with Physician's Choice Treatment or Placebo (Supplementary Fig. 9b). As the reported incidences so far are low and incomprehensive, and bone pain is common in metastatic cancer patients, these interesting observations will require further comprehensive clinical evaluation.

## Discussion

Together, our findings suggest that PARP1/2 dual inhibitors may increase cancer bone metastasis through PARP2-dependent regulation of IMCs, and cause bone loss through PARP1/2-dependent regulation of osteoclasts (Fig. 4i). Many BRCA-mutated cancer patients do not respond to PARP inhibitor treatment. Acquired PARP inhibitor resistance is commonly observed due to multiple mechanisms, including BRCA function restoration by secondary mutations[26]. Our findings that PARP1/2 dual inhibitors may lead to bone loss and exacerbate bone metastasis further emphasize the importance of effective biomarkers to predict the sensitivity of PARP inhibitors and the long-term clinical outcome. Currently, PARP inhibitors are only approved for treating BRCA-mutated breast cancer and ovarian cancer patients. Given our findings, it is critical to carefully examine whether PARP inhibitor treatment triggers metastatic bone diseases in patients with restored BRCA functions and/or acquired PARP inhibitor resistance. Furthermore, it is important to assess whether patients receiving PARP inhibitor treatment may have reduced bone mass and increased fracture risk. Drugs suppressing osteoclast activity, such as RANKL antibody and bisphosphonates may attenuate olaparib-induced bone loss. Inhibitors of β-catenin, CCL3 replenishment, anti-RANKL antibody, and bisphosphonates may prevent olaparib-induced bone metastasis.

By establishing innovative experiment tools combining in vitro and in vivo, gain- and loss-of-function, genetic and pharmacological approaches, this study has identified and characterized PARP2 deficiency as a mediator of breast cancer bone metastasis that functions via multiple mechanisms. This study has also uncovered important functional and mechanistic distinctions between PARP1 and PARP2. Our findings warrant potential deleterious effects of current PARP1/2 inhibitor cancer drugs to exacerbate bone metastasis and trigger bone loss; provide key insights for personalized medicine and tailored treatment using PARP1/2 dual inhibitors by revealing distinct effects on bone metastasis, depending on cancer cell BRCA status and PARP inhibitor sensitivity. These findings highlight the importance of future study on developing and evaluating PARP1-specific inhibitors as possibly safer options and PARP2-specific activators as potential strategy to prevent and treat bone metastasis. Our findings provide possibilities to extend PARP inhibitors for the treatment of BRCA-WT breast cancer patients or PARP inhibitor-resistant patients through cotreatment of CCL3, β-catenin inhibitors, anti-RANKL antibody, or bisphosphonates. Collectively, our data unveil a hidden yet critical chapter in cancer biology, PARP biology, and skeletal biology, with significant clinical and therapeutic implications in a variety of diseases, such as cancer and osteoporosis.

## Methods

**Mice**. PARP1 and PARP2 global KO mice, as well as PARP2 flox mice were provided by Dr. Francoise Dantzer[5,19,27]. Generation of PARP1 flox mice has been published[28]. To establish conditional PARP1 or PARP2 KO in the myeloid lineage, PARP1 or PARP2 flox mice were bred with LyzM-Cre mice[29]. MMTV-PyMT transgenic C57BL/6 mice and MMTV-Cre mice were from the Jackson Laboratory. WT C57BL/6 J mice (UTSW Breeding Core), BALB/cJ mice (UTSW Breeding Core), and athymic nude mice (Charles River) were also used. Nude PARP1 and PARP2 global KO mice were established by breeding PARP1 and PARP2 global KO C57BL/6 mice with athymic nude mice. All experiments were conducted using littermates. All protocols for mouse experiments were approved by the Institutional Animal Care and Use Committee of University of Texas Southwestern Medical Center. All mice were housed in accordance with approved IACUC protocols. Animals were housed on a 12–12 light cycle (light on 6:00, off 18:00) and provided food and water ad libitum.

**Reagent**. The TOP-flash luciferase reporter and FOP-flash negative control were from Dr. Chi Zhang (Texas Scottish Rite Hospital for Children). PCR products for human or mouse PARP1/2 and mouse CCL3 were amplified from their cognate complementary DNAs (cDNAs), and cloned into pTy-U6 vector (provided by Yi

Zhang, UT Southwestern Medical Center). PARP2 mutants were constructed via overlap extension PCR. PARP1 shRNA and PARP2 shRNA were generated by cloning double-stranded oligonucleotides into the pLKO.1 vector (Sigma). The shRNA constructs targeting human or mouse BRCA1 mRNA (TRCN0000244984, TRCN0000042558), non-targeting control shRNA (shNT, SHC002), and anti-β-actin antibody were purchased from Sigma-Aldrich (St. Louis, MO). Antibodies for PARP1 and PARP2 were purchased from Active Motif (Carlsbad, CA). Antibodies for β-catenin and p65 were from Santa Cruz Biotechnology (Dallas, TX). Antibody for H3K4me3 was from Abcam (Cambridge, MA). Anti-poly-ADP-ribose binding reagent was from MilliporeSigma (Burlington, MA).

**Cell culture**. HEK293T cells, 4T1.2 cells, and MDA231-BoM-1833 cells were cultured in DMEM medium supplemented with 10% FBS and 1× Antibiotic-Antimycotic (Invitrogen) at 37 °C with 5% $CO_2$. Py8119 cells were cultured in Ham's F12K medium containing 5% FCS (Fetal Clone II), MITO Serum Extender, and 1× Antibiotic-Antimycotic. RAW264.7 mouse macrophage cell line was from ATCC (TIB-71) and was cultured in α-MEM with 10% FBS and 1× Antibiotic-Antimycotic at 37 °C with 5% $CO_2$. All reagents for cell culture were from Fisher Scientific. To generate stable cell lines, the pTy-U6 or pLKO.1 plasmid was cotransfected with the expression plasmids for GAG-Pol-Rev, VSV-G envelope protein, and pAdvantage into HEK293T cells using Lipofectamine 3000 (Invitrogen). The resulting virus was then filtered and used to infect 1833 cells or Py8119 cells followed by puromycin selection.

**Bone analyses**. Bone volume and architecture were evaluated with μCT using a Scanco μCT-35 instrument (SCANCO Medical) as described with minor modifications[30]. Mouse tibiae were fixed in 70% ethanol and scanned for overall tibial assessment (7 μm resolution), and the structural analysis of trabecular (3.5 μm resolution) and cortical bone (7 μm resolution). Trabecular bone parameters were calculated using Scanco software to analyze the bone scans from the trabecular region directly distal to the proximal tibial growth plate. Femurs were fixed in 2% paraformaldehyde for 24 h, decalcified with 10% EDTA (pH 7.5) for 8 days, emerged in 30% sucrose in PBS for 24 h, embedded in optimal cutting temperature compound (Fisher Scientific), and sectioned at 12 μm. TRAP staining of osteoclasts or femur sections was performed using the Leukocyte Acid Phosphatase staining kit (Sigma). Histomorphometric analyses were conducted using the BIOQUANT Image Analysis software (Bioquant Version 14.1.6). As a bone resorption marker, serum CTX-1 was measured with a RatLaps EIA kit (Immunodiagnostic Systems). As a bone formation marker, serum amino-terminal propeptide of type 1 collagen (P1NP) was measured with a Rat/Mouse P1NP enzyme immunoassay kit (Immunodiagnostic Systems). H&E staining was performed by Histology Core of UT Southwestern Medical Center.

**Osteoclast differentiation**. Osteoclasts were differentiated from bone marrow cells as previously described[15,30]. Briefly, hematopoietic bone marrow cells were filtered with a 40 μm cell strainer, and differentiated with 40 ng/mL of mouse M-CSF (R&D Systems, Minneapolis, MN) in α-MEM containing 10% FBS for 3 days, then with 40 ng/mL of mouse M-CSF and 100 ng/mL of mouse RANKL (R&D Systems) for 3–9 days, in the presence or absence of 1 μM rosiglitazone (Cayman Chemical, Ann Arbor, MI). In coculture experiments, after culturing bone marrow cells with M-CSF for 3 days, cancer cells (1/100 amount of seeded bone marrow cells) were added together with RANKL. For inhibitor treatment, indicated dosages of olaparib (LC Laboratories, Woburn, MA) or PDD 00017273 (R&D Systems) were added during the entire differentiation process. Osteoclast differentiation was quantified by the RNA expression of osteoclast marker genes (TRAP and CTSK) using reverse-transcription quantitative PCR, as well as TRAP staining of osteoclasts.

**Gene expression analyses**. RNA was extracted with Trizol (Invitrogen) and reverse transcribed into cDNA using an ABI High Capacity cDNA RT Kit (Invitrogen) and then analyzed using Applied Biosystems 7700 real-time PCR (SYBR Green) in triplicate with qPCR primers (Supplementary Table 2). All RNA expression was normalized by RPL19 (mouse) or GAPDH (human). Protein expression was analyzed by western blot using whole-cell extract with antibodies against PARP1 (1:1000), PARP2 (1:1000), β-actin (1:5000), PAR (1:1000), or β-catenin (1:1000). CCL3 protein levels in conditioned medium were measured with a mouse CCL3 ELISA kit (R&D Systems).

**Bone metastasis analyses**. Using a VisualSonics Vevo 770 small-animal ultrasound device, luciferase-labeled cancer cells were injected into the left cardiac ventricle so that they could bypass the lung and efficiently migrate to the bone[31]. Bone metastases were detected and quantified weekly after injection by BLI using a Caliper Xenogen IVIS Spectrum instrument at University of Texas Southwestern small-animal imaging core facility. All the BLI signal data over the entire course could be found in Supplementary Fig. 10. Mice with massive BLI signals (>$10^{10}$) on chest only were excluded as failed intracardiac injections. The osteolytic metastatic lesions were imaged by radiography using a Faxitron Cabinet X-ray System with the X-ray tube peak kilovoltage fixed at 26 kVp and the exposure time at 15 s. Radiolucent osteolytic lesions in the hind limbs of mice were quantified. The luciferase-labeled bone-metastasis-prone MDA-MB-231 human breast cancer cell

subline (MDA231-BoM-1833)[12] was provided by J. Massague´ and injected into 6-week-old female nude mice (NCI) at $1 \times 10^5$ cells per mouse in 100 μl PBS. The luciferase-labeled 4T1.2 mouse mammary tumor subline[13] was provided by Robin Anderson (Peter MacCallum Cancer Centre) and Yibing Kang (Princeton University), and was injected into 6-week-old female BALB/c mice at $1 \times 10^5$ cells per mouse in 100 μl PBS. The luciferase-labeled Py8119 bone-metastatic-prone mouse mammary tumor cell line[14] originally derived from spontaneous mammary tumors in C57BL/6 MMTV-PyMT female transgenic mice[32] was provided by Dr. Lesley Ellies (UCSD), and injected into 6-week-old female C57BL/6 J mice at $2.5 \times 10^4$ cells per mouse in 100 μl PBS. For in vivo inhibitor treatment, mice were treated with DMSO or 50 mg/kg of olaparib and/or 50 mg/kg of BX-471, 10 mg/kg of Maraviroc (DC chemicals, China), and 20 mg/kg of PRI-724 (Selleckchem, Houston, TX) by i.p. injection daily starting 1 week before tumor cell injection. Anti-mouse RANKL antibody (clone IK22/5, Biolegend) was injected via i.p. at 250 μg/mice on the same day before intracardiac injection. ZOL (Sigma) was i.p. injected at 0.1 mg/kg weekly in week −1, 0, and 1.

**Analyses of primary tumor growth and spontaneous metastases.** Female nude or C57BL/6 mice (6–8 week old) were injected with $5 \times 10^5$ cells into the fourth mammary fat pad. Tumor size was measured every 3 days using electronic calipers. Tumor volume was calculated as (length × width$^2$)/2. The tumor-free survival of MMTV-PyMT mice was calculated as the duration between date of birth and the initial observation of mammary tumors. The time of primary tumor growth was from tumor onset to the maximal tumor volume limit (single tumor 1500 mm$^3$; all tumors 2000 mm$^3$). Spontaneous lung metastases were detected as described[33]. Entire lungs were excised from mice once the tumor volume reached the maximal tumor volume limit. For mice with cancer cell mammary fat pad injection, the genomic DNA was isolated from homogenized tissues. Genomic DNA was subjected to qPCR to detect the cycle threshold (Ct) for vimentin (total mouse tissue) and luciferase (tumor cells). For MMTV-PyMT mice, RNA was isolated from tissues and subjected to detect Ct of RPL19 (total mouse tissue) and PyMT (tumor cells). Relative tumor burden was calculated using $10,000 \times 2^{-\Delta Ct}$, where ΔCt is the difference of the Ct values of vimentin and luciferase or RPL19 and PyMT.

**Promoter analyses.** Genomic DNA fragments from upstream 2 kb, 0.7 kb, 0.5 kb, 0.3 kb, or 0.1 kb to the mouse or human CCL3 transcription start site were cloned into the pGL4.20 vector (a kind gift from Dr. Beth Levine, UT Southwestern Medical Center) to generate the CCL3-luc reporters. In TOP/FOP flash assay, TCF binding luciferase reporter (TOP-flash) or a control reporter with mutant TCF binding sites (FOP flash) were used to quantify β-catenin transcriptional activity. Luciferase reporters were transiently cotransfected with a cytomegalovirus-β-gal reporter (as internal control for transfection efficiency), along with knocking-down plasmids for PARP1 or PARP2 into HEK293T cells using Lipofectamine 3000 (Invitrogen). Luciferase activity was measured after 48 h and normalized by β-gal activity.

**Chromatin immunoprecipitation.** ChIP assays were performed as described[34]. Briefly, cells in 15-cm dishes were cross-linked with 1% formaldehyde, quenched in 137.5 mM glycine, and harvested in PBS. Then cells were incubated in cell lysis buffer (10 mM HEPES pH 8.0, 85 mM KCl, and 0.5% Nonidet P-40) for 10 min on ice. Nuclei were pelleted at $100 \times g$ for 10 min at 4 °C and resuspended in nuclei lysis buffer (50 mM Tris-HCl pH 8.0, 10 mM EDTA, 1% SDS, and proteinase inhibitor cocktail). Samples were sonicated at 40% power for 30 s for three times. We kept 10% supernatant as input and the rest was incubated with 4 μg of antibodies overnight at 4 °C followed by incubation with protein A/G beads for 2 h. Beads were washed with high salt buffer (50 mM HEPES pH 7.9, 500 mM NaCl, 1 mM EDTA, 0.1% SDS, 1% Triton X-100, and 0.1 % deoxycholate) for four times and TE buffer (10 mM Tris-HCl pH 8.0, and 1 mM EDTA) twice. Then the beads were incubated in elution buffer (50 mM Tris-HCl pH 8.0, 10 mM EDTA, 1% SDS, and 66.7 ng/μl proteinase K) for 2 h at 55 °C followed by incubation at 65 °C overnight to reverse cross-link. DNA was then purified with PCR purification kit (Qiagen, Germantown, MD). ChIP output was quantified by qPCR in triplicates and normalized by input. Four pairs of primers CCL3-ChIP-1, 2, 3, and 4 located on +4 to −117, −163 to −285, −348 to −468, and −504 to −627 of CCL3 promoter, respectively, were used to detect the affinities of CCL3 promoter with proteins.

**Flow cytometry analyses.** Bone marrow cells were isolated and filtered with a 100 μm cell strainer. Then cells were treated with ACK (ammonium–chloride–potassium) lysis buffer (150 mM NH$_4$Cl, 10 mM KHCO$_3$, 1 mM EDTA, pH 7.2) for 3 min on ice. Cells were blocked with anti-CD16/32 (anti-Fcγ R III/II receptor, clone 93, 1:1000 dilution) for 20 min and then stained for 20 min with 7-AAD and antibodies (1:200 dilution) against CD45 (clone 30-F11, BD Biosciences), CD11b (clone M1/70), F4/80 (clone BM8), CD11c (clone N418), MHC II (clone M5/114.15.2), Gr1 (clone RB6-8C5), B220 (clone RA3-6B2), NK1.1 (clone PK136), CD3 (clone 17A2), CD4 (clone RM4-5), CD8a (clone 53-6.7), and CD25 (clone PC61). Treg cells were analyzed by further treatment with Foxp3/Transcription Factor Staining Buffer (Invitrogen) and stained with anti-FoxP3 (1:200 dilution, clone FJK-16s, Invitrogen). To analyze Th1 and Th2 cells and Th17 cells, bone marrow cells were stimulated with Cell Activation Cocktail plus Brefeldin A

(Biolegend) for 4–6 h after ACK treatment. After cell surface staining, cells were further treated with Intracellular Fixation and Permeabilization Buffer (Invitrogen) and stained with antibodies (1:200 dilution) against IFNγ (clone XMG1.2), IL-4 (clone 11B11), or IL-17A (clone TC11-18H10.1). Cells were analyzed on LSR II flow cytometer (BD Biosciences) at the flow cytometry core of UT Southwestern Medical Center. Data were collected with the BD FACSDiva software (Version 8.0.1) and analyzed with the BD FlowJo Version 10 software. Gating strategies were presented in Supplementary Fig. 11. All reagent and antibodies were purchased from Biolegend (San Diego, CA) unless specified otherwise.

**RNA-seq.** RNA-seq was performed in the Next-Generation Sequencing Core at UT Southwestern Medical Center. Briefly, total RNA was extracted with Trizol and miRNeasy Mini Kit (Qiagen). Four μg of total DNase-treated RNA were prepared with the TruSeq Stranded Total RNA LT Sample Prep Kit (Illumina). RNA was purified and fragmented before strand-specific cDNA synthesis. cDNA was then A-tailed, ligated with indexed adapters, PCR amplified, and purified with Ampure XP beads. Library quality was validated on the Agilent 2100 Bioanalyzer. Samples were quantified by Qubit before being normalized and pooled, and then run on the Illumina HiSeq 2500 using SBS v3 reagents to obtain single-end 75 bp reads at a depth of 35 million reads per sample. Quality control was performed with FastQC (version 0.11.2) and FastQ Screen (version 0.4.4). Reads were trimmed to remove adapter sequences using fastq-mcf (ea-utils/version 1.1.2-806), then mapped to mm10 genome using TopHat (version 2.0.12) with duplicated reads marked by Picard tools (version 1.127). RNA counts were generated from Subread/Feature-Counts. Differential expression analysis of FPKM (Fragments Per Kilobase of exon model per Million mapped fragments) was carried out by edgeR. Differential expressed genes were determined by false discovery rate <0.05.

**Cell proliferation, migration, and clonogenic assay.** Cells were seeded into 96-well plates at a density of 2000 cells per well, and cell proliferation was measured by testing luciferase activity on days 1, 3, 5, and 7. Cancer cell migration was quantified using corning transwell plates (8 μm pore size, Fisher scientific). Briefly, $5 \times 10^5$ cells in 100 μL serum-free medium were seeded in the upper chamber and 600 μL medium containing 10% FBS was placed in the bottom chamber. After 24 or 48 h, the relative number of cells that migrated to the bottom chamber was quantified by measuring the luciferase activity. In CCL3 chemotaxis assay, $1 \times 10^6$ bone marrow cells (ACK buffer treated) in 100 μL culture medium were seeded in the upper chamber of transwell plates (5 μm pore size). Culture medium (600 μL) with or without 200 ng/mL recombinant mouse CCL3 (Biolegend), 1 μM CCR1 inhibitor BX-471 (Millipore), or 1 μM CCR5 inhibitor Maraviroc (Fisher scientific) was placed in the bottom chamber. The plates were incubated at 37 °C for 2 h, and then the cells in the bottom chamber were counted and collected for FACS analysis. In the clonogenic assay, 200 cells were plated in each well of six-well plates and cultured for 2 weeks. The cell colonies were fixed and stained in 4% formaldehyde containing 0.5% (w/v) crystal violet for 30 min. The size of colonies was analyzed using the ImageJ software (version 1.4.3.67).

**In vitro PARylation assay.** In each reaction, 500 ng recombinant GST-tagged human β-catenin (sigma) was mixed with 500 ng recombinant human PARP1 (Tulip Biolabs) or PARP2 (Sigma) together with or without 10 μM olaparib in 100 μL reaction buffer (50 mM Tris pH 7.5, 4 mM MgCl$_2$, 20 mM NaCl, 250 μM DTT, 100 ng sheared salmon DNA (ambion), and 500 μM β-NAD$^+$ (Sigma)). The mixture was incubated at room temperature for 1 h. The reactions were terminated by SDS loading buffer and subjected to western blot.

**Statistical analyses.** Statistical parameters including the definitions and exact value of number, deviations, P values, and the types of the statistical tests are reported in the figures and figure legends. We performed all statistical analyses in GraphPad Prism version 7 software with data from three or more biologically independent replicates using two-sided Student's t-test unless noted otherwise. Statistical analysis of primary tumor growth was performed using two-way ANOVA with Sidak's multiple comparison. Two-sided log-rank test was used to analyze tumor-free survival. Two-sided Mann–Whitney U test was performed for analyzing tumor burden (non-normally distributed data). Results are represented as mean ± s.e.m. The P values are *$P < 0.05$, **$P < 0.01$, ***$P < 0.001$, and ****$P < 0.0001$; NS, nonsignificant ($P > 0.05$).

**Reporting summary.** Further information on research design is available in the Nature Research Reporting Summary linked to this article.

## Data availability

The RNA-seq data have been deposited in the Gene Expression Omnibus database under the accession code GSE145755. Clinical data are available in Clinicaltrials.gov with Identifier NCT00494234, NCT00494442, NCT00628251, NCT00753545, NCT00679783, NCT01078662, NCT02000622, NCT01844986, NCT01874353, NCT01945775, NCT02034916, NCT01891344, NCT01968213, and NCT01847274. The source data underlying Figs. 1a–f, 2b–d, f–h, 3a–c, h–n, and 4a–h, and Supplementary Figs. 1a, 1c–n, p, q, 2a–p, 3a, b, d–f, h–k, 4a–f, h, 5c–g, 6a–g, 7a–h, j, k, and 8a–h are provided as a

Source Data file. All the other data supporting, the findings of this study are available within the article, and its supplementary information files and from the corresponding author upon reasonable request. A reporting summary for this article is available as a Supplementary Information file.

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

## Acknowledgements

We thank University of Texas Southwestern histology pathology core, flow cytometry core, and next-generation sequencing core for their assistance in our studies; F. Dantzer and W. L. Kraus for the PARP1 and PARP2 KO, and flox mice; W. L. Kraus for the plasmids for human and mouse PARP1 and PARP2, and shPARP1 and shPARP2; S. Morrison, P. Dechow, J. Feng, and C. Qin for assistance with μCT, histomorphometry, and X-ray analysis. We also thank University of Texas Southwestern Small-Animal Imaging Resource, which is supported in part by the Harold C. Simmons Cancer Center through an NCI Cancer Center Support Grant, 1P30 CA142543-01 and The Department of Radiology. Y.W. is Lawrence Raisz Professor in Bone Cell Metabolism and a Virginia Murchison Linthicum Scholar in Medical Research. This work was in part supported by NIH/NCI (R01CA229487, R01CA236802, Y.W.), CPRIT (RP180047, Y.W.), and DOD (W81XWH-18-1-0014, Y.W.), The Welch Foundation (I-1751, Y.W.) and UTSW Endowed Scholar Startup Fund (Y.W.). The VisualSonics Vevo 770 was purchased with National Institutes of Health American Recovery and Reinvestment Act stimulus funds 1S10RR02564801.

## Author contributions

H.Z. and Y.W. conceived the project and designed the experiments. All experiments, except those listed below, were performed by H.Z. D.Y., Q.Y., H.T., and Y.-X.F. assisted with FACS. H.Z. and Y.W. wrote the manuscript.

## Competing interests

The authors declare no competing interests.
