## [Peer Review File · Nature Communications]

Reviewers' Comments:

Reviewer #2:

Remarks to the Author:

The manuscript by Wan and colleagues highlights so far uncharacterized, PARP2 specific functions in the bone and reveals alterations in the bone structure using olaparib treatment, which is first- or second-line treatment for several cancer types. This data is very interesting and could be of high relevance.

The authors have substantially revised their manuscript and added additional data to substantiate their findings. The clarity has improved and the manuscript now reads well. All of my concerns were addressed.

Reviewer #3:

Remarks to the Author:

I have read the manuscript and the detailed responses to the previous 3 reviewers. The authors have made substantial changes.

I would caution on over interpreting the clinical data. In supplementary table 9a, that data does not on appearance really demonstrate significant differences between PCT chemotherapy and the PARP inhibitor- as bone pain etc. is very common in metastatic patients. Consider adding N/A to those areas where there was no comparison arm so that readers understand. It would not be standard practice to follow or gather future scans and evaluate them on patients in trials when they come off of study so it would need to be collected prospectively.

I think the only thing you can say clinically is preclinical is interesting and should be evaluated in the clinical setting.

Reviewer #4:

Remarks to the Author:

Response to Reviewer 1

1. Concerns regarding the use of cell lines. This is somewhat addressed. The work is strengthened by the inclusion of the 4T1 data, providing 3 in vivo models that demonstrate increased bone mets in response to Olaparib treatment. There remains a great deal of confusion with the cell lines, as the knockout mouse experiments appear to be done using 8119 cells whereas the knockout cells are using 1833, which makes it very difficult to compare.

2. Clinical data. This was raised by both reviewers, and has not really been adequately addressed, however this is because the data is unavailable. The authors have presented some studies that suggest a link between PARP inhibition and bone loss, but it remains difficult to know whether this effect of PARP inhibition will translate to the clinic or if it is an artefact of preclinical models.

3. Questions about CD11b/Gr1 and other populations. The authors have performed additional experiments which partly address the question

4. The question regarding CCL3 has been appropriately addressed

5. The remaining questions have been addressed

Additional Comments

1. In my opinion, the bone metastasis experiments have not been performed to an appropriate standard. The data is merely microCT to quantitate trabecular bone volume and luciferase for tumour burden, with CTX as a marker of bone resorption. For a manuscript which proposes a new mechanism by which PARP inhibition alters osteoclast biology it is not acceptable that osteoclasts and osteolytic lesions have not been quantitated. Osteoblasts should also be quantitated, and one would assume that RANKL expression would be altered as these are the major source of RANKL within the bone microenvironment.

2. Unless I have missed it, the authors have not demonstrated a direct effect of Olaparib on osteoclast differentiation and bone resorption in vitro, merely changes in ex vivo PARP 1 and 2 knockdown cells. Given the translational significance of the findings, it is key to show that a clinically relevant PARP inhibitor has direct effects to increase osteoclast formation and activity.

We sincerely thank the reviewers for their valuable suggestions and their effort on improving our manuscript. We have revised the text (changes underlined) and figures to address all the comments as detailed below. We hope that these revisions will satisfy the reviewers and this manuscript will now be accepted for publication.

Reviewer #2 (Remarks to the Author):

The manuscript by Wan and colleagues highlights so far uncharacterized, PARP2 specific functions in the bone and reveals alterations in the bone structure using olaparib treatment, which is first- or second-line treatment for several cancer types. This data is very interesting and could be of high relevance. The authors have substantially revised their manuscript and added additional data to substantiate their findings. The clarity has improved and the manuscript now reads well. All of my concerns were addressed.

Reviewer #3 (Remarks to the Author):

I have read the manuscript and the detailed responses to the previous 3 reviewers. The authors have made substantial changes. I would caution on over interpreting the clinical data. In supplementary table 9a, that data does not on appearance really demonstrate significant differences between PCT chemotherapy and the PARP inhibitor- as bone pain etc. is very common in metastatic patients. Consider adding N/A to those areas where there was no comparison arm so that readers understand. It would not be standard practice to follow or gather future scans and evaluate them on patients in trials when they come off of study so it would need to be collected prospectively. I think the only thing you can say clinically is preclinical is interesting and should be evaluated in the clinical setting.

Thank you for suggestions. We agree that we should avoid over interpreting the clinical data presented in our manuscript. We have added “n/a” to the blank areas in **Supplementary Fig. 9a**. We have also modified the last paragraph of Results on **Page 13** to clarify the analysis of these data and emphasize the requirement of future clinical evaluation.

Reviewer #4 (Replacement for Reviewer#1, Remarks to the Author):

Response to Reviewer 1

1. Concerns regarding the use of cell lines. This is somewhat addressed. The work is strengthened by the inclusion of the 4T1 data, providing 3 in vivo models that demonstrate increased bone mets in response to Olaparib treatment. There remains a great deal of confusion with the cell lines, as the knockout mouse experiments appear to be done using 8119 cells whereas the knockout cells are using 1833, which makes it very difficult to compare.

We used both 1833 and Py8119 cells for both knockout mouse experiments and the knockdown cells.

1833 cells - nude mice for testing knockdown cells: **Fig. 1c, S1d, S1i**.

1833 cells - nude mice for testing knockout mouse: **Fig. S2a, S2b**.

Py8119 cells - B6 mice for testing knockdown cells: **Fig. S1h, S1j**.

Py8119 cells - B6 mice for testing knockout mouse: **Fig. 1d, 1e, S2c, S2d**.

We also mentioned the cell lines and mouse background in the main text.

2. Clinical data. This was raised by both reviewers, and has not really been adequately addressed, however this is because the data is unavailable. The authors have presented some studies that suggest a link between PARP inhibition and bone loss, but it remains difficult to know whether this effect of PARP inhibition will translate to the clinic or if it is an artifact of preclinical models.

We thank the reviewer for the suggestions. We have now clarified that our analysis is not comprehensive and further clinical evaluation is needed on **Page 13**.

3. Questions about CD11b/Gr1 and other populations. The authors have performed additional experiments which partly address the question

4. The question regarding CCL3 has been appropriately addressed

5. The remaining questions have been addressed

Additional Comments

1. In my opinion, the bone metastasis experiments have not been performed to an appropriate standard. The data is merely microCT to quantitate trabecular bone volume and luciferase for tumor burden, with CTX as a marker of bone resorption. For a manuscript which proposes a new mechanism by which PARP inhibition alters osteoclast biology it is not acceptable that osteoclasts and osteolytic lesions have not been quantitated. Osteoblasts should also be quantitated, and one would assume that RANKL expression would be altered as these are the major source of RANKL within the bone microenvironment.

Thank you for the suggestions. The osteoclast quantitation has been performed as shown in **Fig. 2c-d, 2g-h, S3e-f, S3i-j**. The effect of PARP inhibitor on osteoblasts has been reported. We have now added the reference and our data showing that olaparib, PARP1KO and PARP2KO reduce bone formation on **Page 8** and in a new figure (**Fig. S3k**). We have also tested RANKL expression in osteoblasts. **As shown in the figure below**, RANKL expression in osteoblasts was not affected by PARP1KO or PARP2KO.

In bone metastasis experiments, H&E staining (**Fig. S1b**) showed tumor cell growth in bone and confirmed luciferase data for tumor burden. X-ray imaging (**Fig. S1a, S1e, S1g**) illustrated the osteolytic lesions in bone metastasis. The increased serum CTX-1 levels in both nude and B6 mice with bone metastasis (**Fig. S1k, S1l**) demonstrated elevated bone resorption.

Figure legend: Osteoblasts were differentiated from bone marrow MSCs. Primary bone marrow cells were cultured for 4 days in MSC medium (Mouse MesenCult Proliferation Kit; StemCell Technologies), then differentiated in α -MEM containing 10% FBS, 5 mM β -glycerophosphate, and 100 μ g/mL ascorbic acid (AA, mineralization medium) (GPAA mixture) for 9 days. The expression of RANKL was not affected by PARP1KO or PARP2KO in osteoblasts (+AA groups).

2. Unless I have missed it, the authors have not demonstrated a direct effect of Olaparib on osteoclast differentiation and bone resorption in vitro, merely changes in ex vivo PARP 1 and 2 knockdown cells. Given the translational significance of the findings, it is key to show that a clinically relevant PARP inhibitor has direct effects to increase osteoclast formation and activity.

Thank you for the comments. As shown in **Fig. 2a-d**, we have indeed tested the direct effect of olaparib on osteoclast and bone both *in vitro* and *in vivo*. **Fig. 2a-b** showed trabecular bone from mice treated with DMSO or olaparib. **Fig. S3b** showed the effect of olaparib on cortical bone. **Fig. 2c** showed osteoclast quantitation in bone histology. **Fig. 2d** showed the direct effect of olaparib on promoting osteoclast differentiation from primary bone marrow cell culture.

We thank all the reviewers for their comments and suggestions.

Reviewers' Comments:

Reviewer #4:

Remarks to the Author:

The authors have not really adequately addressed my concerns about their assessment of bone disease in the bone metastasis model. The quantitation they refer to is all in the knockout models, not the bone metastatic models. The H&E and X-ray they refer to is one representative image, with no attempt at quantitation. While this is disappointing, it does not detract from the overall results or the importance of the study.

They have addressed all other concerns.

Reviewer #4 (Remarks to the Author):

The authors have not really adequately addressed my concerns about their assessment of bone disease in the bone metastasis model. The quantitation they refer to is all in the knockout models, not the bone metastatic models. The H&E and X-ray they refer to is one representative image, with no attempt at quantitation. While this is disappointing, it does not detract from the overall results or the importance of the study.

Thank you for the comments. We concur that it is important to quantify bone disease in the bone metastasis model. We have indeed collected all X-ray images for bone metastasis experiments. Now in addition to showing the representative X-ray images, we have also included the quantification of radiolucent osteolytic lesions in the hind limbs of mice in the bone metastasis model (**Supplementary Fig. 1a, e, g**). Raw data for the quantitation in these figures are included in the Data Source file, and one example is shown below. Because we used most of the mice (especially the hind limbs) for X-ray or FACS analyses, we did not analyze all mice with H&E staining quantitation; we only performed H&E staining on a subset of mice to confirm the presence of bone metastases at the bioluminescence indicated sites.